# The Method of Creel Positioning Based on Monocular Vision

**DOI:** 10.3390/s22176657

**Published:** 2022-09-02

**Authors:** Jiajia Tu, Sijie Han, Lei Sun, Weimin Shi, Ning Dai

**Affiliations:** 1School of Automation, Zhejiang Institute of Mechanical & Electrical Engineering, Hangzhou 310053, China; 2School of Mechanical Engineering & Automation, Zhejiang Sci-Tech University, Hangzhou 310018, China

**Keywords:** monocular vision, yarn frame, creel, visual positioning, image processing

## Abstract

Automatic replacement of bobbins on the frame is one of the core problems that must be solved in the textile industry. In the process of changing the bobbin, it is faced with problems such as the position offset of the bobbin, the loosening and deformation of the bobbin, which will lead to the failure of replacing the bobbin. Therefore, it is necessary to initialize the creel coordinates regularly, also considering the labor intensity and poor reliability of manual positioning. This paper proposes an automatic creel positioning method based on monocular vision. Firstly, the industrial camera was installed on the drum changing manipulator controlled by the truss system, and each yarn frame in the yarn area was inspected. Secondly, the end face image of the creel was collected at a reasonable distance, and the collected images were transmitted to the computer in real time through the Ethernet bus. Thirdly, the center coordinates (x, y) and radius r of the creel were marked by the improved Hough circle detection algorithm. Finally, the coordinate deviation of the creel was calculated and transmitted to the system controller to realize deviation correction. Before the test, the creel positioning markers were specially designed and the camera was calibrated. Thus, the influence of image complex background, creel end roughness, reflection and other factors can be reduced, and the image processing speed and positioning accuracy can be improved. The results show that the positioning effect of this method is fine when the distance between the center of the camera lens and the center of the end face of the creel is 170~190 mm. Moreover, when the distance is 190 mm, the positioning effect is the best, with an average error of only 0.51 mm. In addition, the deviation between the center coordinate and the radius of the end face of the marker is also very small, which is better than the requirements of bobbin yarn changing accuracy.

## 1. Introduction

The floor frame is a form of knitting “weft knitting round machine” frame. The frame is divided into two rows and stands on both sides of the machine in a linear or circular shape, and the bobbins are inserted obliquely on the creel of the frame. Although the floor yarn frame covers a large area and has a long yarn supply distance, it has the advantages of small machine load-bearing and convenient yarn change [1]. At present, it is widely used in the production workshop of circular weft knitting machine, and the yarn bobbin yarn is replaced manually in the traditional way for thread continuation. With the deepening of problems such as high labor cost, high labor intensity and difficult recruitment, it has begun to affect the normal development of the textile industry, and enterprises have put forward an urgent need for “machine replacement” [2,3,4]. Therefore, in the context of Intelligent Manufacturing, it is an inevitable trend to use robots to automatically replace the bobbin on the yarn frame [5].

Although the positioning accuracy required by the manipulator can be achieved through the cooperation of the truss mechanism and the servo system when the machine is changing bobbins, there are many problems such as the low manufacturing accuracy of the creel, the looseness of the creel, the inevitable vibration in the process of automatic drum changing, the displacement of the creel, and the sag of the creel caused by the heavy weight of the creel. All of these will affect the normal bobbin replacement, so the coordinates of the creel must be regularly initialized to ensure the manipulator can replace the bobbin normally. However, the manual initialization and positioning of the creel coordinates has a large workload and poor reliability, so it is considered to realize the initialization and positioning of the creel coordinates based on machine vision.

At present, there is a lot of research on visual positioning, such as the accuracy analysis of threaded hole positioning [6], research on robot arm positioning method [7], and object pose measurement and error [8], but studies on the textile industry are relatively few. The reason is that textiles, as a traditional labor-intensive industry, has not paid enough attention to automation and intelligence in the past, and started relatively late. In addition, the textile factory environment is complex, and there are too many background contents of image acquisition, which leads to difficulties in image processing, and the high cost of vision system investment, which is difficult for enterprises to bear [9]. Therefore, based on the existing results of monocular vision positioning, this paper will study the system and algorithm suitable for the coordinate positioning of creels, mainly including principal analysis, system design, algorithm design, experimental testing and other aspects, so as to provide help for the early realization of automatic drum changing of creels and intelligent manufacturing in the textile industry.

In the production workshop of knitting circular weft knitting machine, the structure diagram of the commonly used floor frame is shown in Figure 1. It is composed of two rows of eight-layer creels on the left and right, and a simple support. The creels on the same layer are placed with the same specification of bobbins, which can realize continuous yarn supply without stopping. In this paper, machine vision technology will be used to locate the creel to meet the requirements of “machine replacing bobbin”.

Since the diameter of the end face of the creel is known, the unknown variables of positioning in image processing are reduced. Then, by designing markers, integrating circle detection and monocular vision technology, the recognition and positioning function of the creel can be realized simply and quickly.

The rest of the paper is organized as follows: Section 2 is the related works. Section 3 presents the prepare works. Section 4 is the main part of this paper; it proposes two steps to detect the creel. Section 5 provides the experiential results and analysis of the method. Section 6 presents the conclusion and discussion of this study.

## 2. Related Works

According to our investigation, the replacement of bobbins on the yarn frame of circular weft knitting machine is still in the pure manual stage at present. There is no research on the positioning of creels, and there is no method or system that can be directly used to locate creels. In the previous research, the commonly used object position detection methods were laser radar and computer visual positioning technology. Laser radar has the advantages of high measurement accuracy and convenient installation. It is widely used in positioning systems, especially in mobile robots. However, there are relatively few applications in the textile industry. It seems that it is only used in AGV dropping system. Computer vision positioning has become popular in recent years, and applications in all walks of life are very popular. There are also many applications in the textile field, including yarn detection, empty bobbin detection, bobbin color and defect detection, etc., but the textile industry has few applications in positioning. At present, the main application is the positioning of the bobbin when dyeing.

### 2.1. Laser Radar

Laser radar constantly transmits signals, when the signal reaches the surface of the object, it generates reflection, and then receives the reflected signal of the object, so as to realize the positioning and recognition of the object, which is mainly used for navigation and obstacle avoidance. Moreover, the accuracy, feedback speed, anti-jamming ability and effective range of laser radar are significantly better than those of infrared and ultrasonic. The disadvantage is that a single beam of laser radar can scan one face data (LDS) through rotation and cannot complete the perception of the three-dimensional world. In the textile industry, laser radar is mainly used in AGV, which is used to transport packages and cloth, so as to ensure the safety and reliability of AGV in the workshop operation.

Laser radar is widely used in industrial fields, such as ranging, scanning, navigation, obstacle avoidance, etc. [10,11,12,13,14]. In the textile industry, laser radar is currently mainly used in AGV for the handling of raw materials, semi-finished products and finished products in textile workshops, including cotton bobbin yarns, bobbins, cloth, etc. For example, the cotton bobbin yarn handling system based on AGV designed by Chen Xiang, Yuan Hongbing, and Yao Wanli realized the automatic handling of cotton bobbin yarns between carding machine and drawing frame by AGV [15]. Guangdong Yida knitting factory had solved the storage and transportation problems in production by using AGV technology and also improved the freight transfer efficiency of the production line [16]. Duan Bo, Yu Lianqing, and Fan Fei designed the control system of AGV dropping trolley with super long body, and improved the dynamic and static stability of the system by using fuzzy adaptive PID controller [17]. Wang Yu, Shen Danfeng, Wang Rongjun, Li Yaojie, and Li Jingyu studied the trajectory tracking control algorithm of the automatic distribution vehicle and reduced the tracking error and random interference of the distribution AGV in the complex environment through the model predictive control method [18].

### 2.2. Computer Visual

Machine vision is widely used in the field of positioning and measurement. Monocular and binocular vision positioning systems are commonly used at present [8]. Monocular vision localization first uses a single camera to collect the initial image of the target object, and then obtains the spatial position information of the target object through image processing and operation. Therefore, compared with binocular vision, monocular vision is more concise in the system structure, more efficient in image processing, and lower in equipment cost [9,10]. Since the diameter of the creel is known, the unknown variables of positioning in image processing are reduced. Then, by designing markers, the recognition and positioning function of the creel can be realized only by using monocular vision system.

Pinhole imaging principle is used in monocular vision positioning technology to determine the position of objects. Vision positioning is one of the most popular research directions at present and has many applications in the industrial field. For example, Zhou Jiaqi, Shi Lun, and Meng Fanyu combined with the plane positioning scene widely used in monocular vision, proposed a method of rapid camera calibration and depth measurement based on servo mechanism, and realized high-precision and high stability depth direction measurement integrated on the fixture change AGV [19]. Luo Zai, Cai Zeliang, Jiang Wensong, and Yang Li proposed a method for measuring the end trajectory of industrial robots based on monocular vision, with a displacement accuracy of 0.315 mm and a rotation accuracy of 0.365°, achieving the purpose of accurately and dynamically measuring the end trajectory of robots [20]. Xie Yushen, Wu Qingcong, Chen Bai, and Jiang Meng proposed a mobile manipulator grasping operation method based on monocular vision. By modifying the grasping error compensation method of hand eye posture, it can quickly and accurately identify the target and effectively reduce the operation error [21].

There is also a lot of research related to the textile industry. For example, Lai Jiawei, He Yuqing, Li Xiaopeng, Liu Mingqi, and Zhai Xiangyang designed a positioning system of the yarn bar latch based on monocular vision. They combined this with image analysis algorithm to calculate the spatial position of the target, the relative position of the target and the manipulator, and finally achieved the capture of the target, with the positioning error within 2 mm [22]. Shi Zhiwei, Shi Weimin, and Wang Junru proposed a yarn allowance detection system based on machine vision, and processed it with a deep learning algorithm, making the edge detection error less than 5.7 mm, which provided help to the intelligent construction of textile workshops [23]. Wang Wensheng, Li Tianjian, Ran Yuchen, Lu Ying, and Huang Min proposed an improved frequency tuning significance detection algorithm to realize the positioning detection of the bobbin cage bar [24]. Zhang Jianxin and Li Qi established a cheese correction model based on perspective projection theory and realized the online detection of cheese density using machine vision [25]. Jing Junfeng and Guo Gen used machine vision to detect the hairiness of silk cakes by finding the convex hull position in the hairiness contour point, effectively positioning and counting the hairiness of silk cakes [26].

## 3. Preparations

Before starting, there are two tasks that need to be performed. The first one is to set up the creel test system. The second one is to design the creel positioning markers. The details are as follows.

### 3.1. System Design

The creel positioning system is mainly composed of a creel yarn changing manipulator, an industrial camera, a creel rod, an industrial computer and other equipment [27,28]. The industrial camera is installed on the special platform of the bobbin yarn changing manipulator, with good field of vision and no interference. The bobbin yarn changing manipulator is suspended on the truss robot and can follow the truss to move to any position of the yarn frame in the yarn using area. Moreover, it is used to realize the function of grasping and placing the yarn bobbin yarn. The industrial computer is configured at a compromise position near the outside of the yarn using area and the yarn preparation area. Ethernet communication is used with industrial cameras, trusses and manipulator controllers. The main hardware composition of the creel positioning mechanism is shown in Figure 2.

The composition block diagram of creel positioning system is shown in Figure 3, which includes image acquisition, image processing and positioning and rectification.

The positioning steps of creel are as follows. First, according to the instructions issued by the industrial computer, the bobbin yarn changing manipulator is controlled by the delta AS320T to reach the position of the creel that needs to be positioned in the yarn area. Then, the industrial camera is awakened by the industrial computer to capture the creel image, and the image is processed by the industrial computer positioning algorithm to obtain the new creel position coordinates. Finally, it is stored in the database to replace the original coordinate data, which can be called for subsequent bobbin yarn replacement, and updated when the next relocation, so as to ensure the accuracy of bobbin yarn positioning and the success rate of barrel replacement.

### 3.2. Markers Design

There are the following problems in image recognition of the end face of the bobbin directly: the diameter of the end face is only about 8 mm, the color is silver-white, and the process is slightly rough. If the image of the creel is taken directly with an industrial camera for processing, recognition and positioning, the configuration requirements of industrial computers and other equipment will be very high, which will greatly increase the use cost [29,30]. Therefore, the method of setting markers on the creel is adopted to highlight the characteristics of the creel. Figure 4a,b show the visual positioning markers designed or selected by the team. The disc marker has its own contrast, which makes recognition easier and greatly reduces the requirements of hardware configuration. The cover marker is installed on the creel, which is conducive to improving the flatness of the end face and improving the recognition. Its diameter is 10 mm.

By analyzing the layout of the yarn frame in the yarn area and the structure of the yarn frame itself, we determined that the better installation positions of the markers are the support under the creel and the end face of the creel, as shown in Figure 5. The disc marker is directly pasted on the support under the creel, and the object is shown in Figure 5a. Its test recognition effect is fine, but it has high requirements for the camera installation mechanism, which is easy to interfere with. Moreover, due to the uneven end face of the creel, the direct pasting firmness and durability are poor, so it is only suitable for experimental testing. Moreover, since the image processing effect of the end face of the creel does not meet the actual requirements, it is considered to install the sleeve on the creel, as shown in Figure 5b. Moreover, it has the advantages of convenient installation, good durability, wide adaptability of yarn frame type, and the test recognition effect and shooting distance can also meet the requirements. So, it is a good choice for practical application. At the same time, in order to enhance the effect of image processing, we design the end face of the cover as a reflective surface on the basis of Figure 5b, and the actual effect is shown in Figure 5c.

## 4. Proposed Methods

In this part, we use two steps to realize the computer vision positioning creel. The first step is the circle detection based on Hough transform, and the second part is the monocular positioning based on industrial camera. The details are as follows.

### 4.1. Creel Recognition Based on Circle Detection

Hough circle transformation is the process of transforming a circle in the two-dimensional image space into a point in the three-dimensional parameter space, which determined by the radius *r* of the circle and the coordinates of the center (*x*, *y*). Therefore, the circle determined by any three points on the circumference corresponds to a point in the three-dimensional parameter space after Hough transformation [31,32].

The basic idea of Hough circle transformation is to think that every non-zero pixel on the image may be a point on a potential circle. Similarly the Hough line transformation, the Hough circle transformation also generates the cumulative coordinate plane by voting, setting a cumulative weight to locate the circle.

We know that in the Cartesian coordinate system, the equation of the circle is shown as Formula (1).
(1)$$(x-a){}^{2}+(y-b){}^{2}=r^{2}$$

(*x*, *y*) is the point on the circumference, (*a*, *b*) is the center coordinate, and *r* is the radius, which can also be expressed as Formula (2).
(2)$$\{\begin{array}{c} x=a+r\cos\theta \\ y=b+r\sin\theta \end{array}$$

θ is the angle between the line connecting the center (*a*, *b*) and the point (*x*, *y*) on the circle and the horizontal line passing through the center (*a*, *b*). By sorting out Formula (2), we can obtain the following Formula (3).
(3)$$\{\begin{array}{c} a=x-r\cos\theta \\ b=y-r\sin\theta \end{array}$$

Therefore, in the three-dimensional coordinate system composed of *a*, *b* and *r*, a point can uniquely determine a circle.

In Cartesian *x*, *y* coordinate system, all circles passing through a certain point are mapped to *a*, *b* and *r* coordinate system, which is a three-dimensional curve. All circles passing through all non-zero pixels in the *x*, *y* coordinate system constitute many three-dimensional curves in the *a*, *b* and *r* coordinate system.

In the *x* and *y* coordinate systems, there is only one circular equation for all points on the same circle, which maps to the same point in the *a*, *b* and *r* coordinate systems. Therefore, in the *a*, *b*, *r* coordinate system, this point should have N0 curves intersecting the total pixels of the circle. Then, by judging the number of intersections of each point in *a*, *b* and *r*, the point which is greater than a certain threshold is regarded as a circle.

The above is the implementation algorithm of standard Hough circle transformation, but the problem is that its cumulative surface is three-dimensional space, which means that it requires more computational consumption than Hough line transformation. Therefore, OpenCV optimizes the operation of Standard Hough circle transformation. It adopts the “Hough gradient method” detection idea, which is to traverse and accumulate the center of all non-zero points. Considering that the center of the circle must be on the module vector of each point on the circle, that is, on the vertical line perpendicular to the point and passing through the tangent of the point, the intersection of the module vectors on these circles is the center of the circle.

The schematic diagram of the relationship between the module vector and the center of the circle is shown in Figure 6. The Hough gradient method is to find these centers, and then determine them according to the number of intersections of the modulus vector on the “center” and the appropriate threshold value. As shown in Figure 6, if we calculate the gradient of a circle, the gradient direction of all points on the circle faces the center of the circle.

Therefore, the steps of circle detection are as follows. Firstly, the Canny algorithm is used to detect the edge of the image. Secondly, the gradient of all pixels is calculated using Sobel operator. Thirdly, after Canny traverses all non-zero pixel points, it will draw lines along the gradient direction. Each point has an accumulator, when a line passes through the point, the accumulator adds 1. Then, all of the accumulated values are sorted, and all possible center points are found according to the threshold value. Fourth, we calculated the distance between all non-zero pixels in canny image and the center of the circle, sort the distance from small to large, and select the appropriate radius. Fifth, we set the accumulator for the selected radius, and add 1 to the accumulator value that meets the radius.

Considering the factor of efficiency, the Hough circle detection implemented in OpenCV is based on image gradient. The first step is to detect the edge, the second step is to find the possible center of the circle and calculate the optimal radius from the candidate center based on the first step. In OpenCV, some codes and analysis of Hough circle detection algorithm based on image gradient are as follows (Algorithm 1).
**Algorithm 1.** The process of Hough circle detection algorithm based on image gradient1: HoughCircles;2: InputArray image;3: OutputArray circles;4: Int method;5: Double dp;6: Double mindist;7: Double param1;8: Double param2;9: Int minradius;10: Int maxradius;

“InputArray” indicates input image. The data type is generally mat type. It needs to be an 8-bit single channel gray-scale image.

“OutputArray” stores the output vector of the detected circle.

“Method indicates” the detection method used. At present, only the Hough gradient method is available in OpenCV. Fill in Hough for this parameter_grade is enough.

“dp” is double type data, which refers to the reciprocal of the ratio of the center accumulator to the resolution of the input image. For example, if dp = 1, the accumulator and the input image have the same resolution. If dp = 2, the accumulator has half the width and height of the input image.

“Mindist” is the minimum distance between the centers of circles detected by Hough transform.

“Param1” is the corresponding parameter of the detection method set by the third parameter method. For the current only method, the Hough gradient method CV_ HOUGH_Gradient, which represents the high threshold passed to Canny edge detection operator, while the low threshold is half of the high threshold.

“Param2” is also the corresponding parameter of the detection method set by the third parameter method. For the current only method, Hough gradient method_ Gradient, which represents the accumulator threshold of the center of the circle in the detection phase. The smaller it is, the more non-existent circles can be detected, and the larger it is, the more perfect circles can be detected.

“Minradius” indicates the minimum value of the circle radius.

“Maxradius” indicates the maximum value of the circle radius.

Figure 5a is processed by the above algorithm, and the recognition effect of disc markers is shown in Figure 7. It can be seen from the Figure that this algorithm not only realizes the rapid recognition of the marker, but also accurately locates its center. Therefore, it is feasible to use this method to locate the creel.

### 4.2. Creel Positioning Based on Monocular Vision

The schematic diagram of creel positioning based on monocular vision is shown in Figure 8. *O_w_* − *X_w_**Y_w_**Z_w_*, *O**_c_* − *X_c_**Y_c_**Z_c_*, and *O* − *XY* are world coordinate system, camera coordinate system and image coordinate system, u, v is the pixel coordinate system. *O_w_* is the origin position for establishing the world coordinate system. *O**_c_* is the optical center of the camera; P (*X**_w_*, *Y**_w_*, *Z**_w_*) is a point in the world coordinate system. p (*x*, *y*) is the imaging point in the image, the coordinates in the image coordinate system (*x*, *y*), and the coordinates in the pixel coordinate system (*u*, *v*), *f* is the focal length of the camera, equal to the distance between point *O* and *O**_c_* [33,34].

We select an image of the creel for processing, and obtain the coordinates of the creel in the image coordinate system as p (*x*, *y*), the coordinates in the camera coordinate system as P (*x_c_*, *y_c_*, *z_c_*), the coordinates in the world coordinate system as P (*x_w_*, *y_w_*, *z_w_*), and the focal length of the lens as f. Therefore, according to the geometric relationship in Figure 8, the calculation formula of the center position of the creel is obtained as follows.
(4)$$x=f\frac{X_{c}}{Z_{c}}$$
(5)$$y=f\frac{Y_{c}}{Z_{c}}$$

The converted camera pinhole model is shown as (6).
(6)$$\left[\begin{array}{c} x\\ y\\ 1 \end{array}\right]=\left[\begin{array}{c} f\\ 0\\ 0 \end{array}\begin{array}{c} \;\;0\\ \;\;f\\ \;\;0 \end{array}\begin{array}{c} \;\;0\\ \;\;0\\ \;\;1 \end{array}\begin{array}{c} \;\;0\\ \;\;0\\ \;\;0 \end{array}\right]\left[\begin{array}{c} x_{c}\\ y_{c}\\ z_{c}\\ 1 \end{array}\right]$$

Convert the image coordinate system to the world coordinate system. The specific formula is shown as (7).
(7)$$\left[\begin{array}{c} x_{c}\\ y_{c}\\ z_{c}\\ 1 \end{array}\right]=\left[\begin{array}{c} R\;\\ 0 \end{array}\;\begin{array}{c} T\\ 1 \end{array}\right]\left[\begin{array}{c} x_{w}\\ y_{w}\\ z_{w}\\ 1 \end{array}\right]$$

$\left[\begin{array}{c} R\;\\ 0 \end{array}\;\begin{array}{c} T\\ 1 \end{array}\right]$ is the camera external parameter matrix; *R* is the rotation matrix and *T* is the translation matrix, which is used to express the relationship between the origin of the world coordinate system and the optical center of the camera.

Next is the conversion from image coordinate system to pixel coordinate system, and the secondary conversion of the object that has been projected on the imaging plane. The specific formula is shown as follows.
(8)$$\left[\begin{array}{c} u\\ v\\ 1 \end{array}\right]=\left[\begin{array}{c} \frac{1}{dx}\;\\ 0\;\\ 0 \end{array}\begin{array}{c} 0\\ \frac{1}{dy}\\ 0 \end{array}\begin{array}{c} \;\;u_{0}\\ \;\;v_{0}\\ 1 \end{array}\right]\left[\begin{array}{c} x\\ y\\ 1 \end{array}\right]$$

dx and dy are pixel sizes. $u_{0}$ and $v_{0}$ are the actual position of the main point.

By transforming the image coordinate system into pixel coordinate system by Formulas (3)–(5), we can obtain the following formula.
(9)$$\left[\begin{array}{c} u\\ v\\ 1 \end{array}\right]=\left[\begin{array}{c} \frac{f}{dx}\\ 0\\ 0 \end{array}\;\begin{array}{c} \;0\\ \;\frac{f}{dy}\\ 0 \end{array}\;\begin{array}{c} \;u_{0}\\ \;v_{0}\\ 1 \end{array}\right]\left[\begin{array}{c} 1\;\;0\;\;0\;\;0\\ 0\;\;1\;\;0\;\;0\\ 0\;\;0\;\;1\;\;0 \end{array}\right]\left[\begin{array}{c} R\;\\ 0 \end{array}\;\begin{array}{c} T\\ 1 \end{array}\right]\left[\begin{array}{c} x_{w}\\ y_{w}\\ z_{w}\\ 1 \end{array}\right]$$

$\frac{f}{dx}$,$\frac{f}{dy}$ are called normalized focal lengths on the *x*-axis and *y*-axis, respectively. $\left[\begin{array}{c} \frac{f}{dx}\\ 0\\ 0 \end{array}\;\begin{array}{c} \;0\\ \;\frac{f}{dy}\\ 0 \end{array}\;\begin{array}{c} \;u_{0}\\ \;v_{0}\\ 1 \end{array}\right]$ is the camera internal parameter matrix. The internal parameter matrix reflects the attributes of the camera itself. Each camera is different and needs calibration to know these parameters. The combination of internal parameter matrix and external parameter matrix is the camera matrix, which establishes the transformation from three-dimensional point homogeneous coordinates to two-dimensional point homogeneous coordinates.

Assume that the distortion free industrial camera is applied to the system. Its resolution is m × n, and target size is a × b. The radius of the creel end face in the image coordinate system is r1, and the actually measured radius is r2. Through measurement, the spatial position of the origin of the camera coordinate system under the manipulator coordinate system is *P(*$x_{1}$*,*
$y_{1}$*,*
$z_{1}$*)*, and through coordinate conversion, the spatial position coordinate of the creel end face under the manipulator coordinate system is *P’(*$x\frac{r2}{r1}$
*+*
$x_{1}$*,*$y\frac{r2}{r1}$
*+*
$y_{1}$*,*$\frac{r2mf}{r1a}$
*+*
$z_{1}$*).*

According to the monocular vision positioning principle, Figure 5a is further processed to obtain the image shown in Figure 9. It can be seen from the Figure that the distance between the camera, the center of the mark and the coordinates of the center of the circle are successfully obtained, which prove the correctness of this method.

## 5. Experimental Test and Analysis

When testing, the image acquisition equipment adopts the MV-CA050-10GM black-and-white industrial camera produced by Hikvision, with a resolution of 2448 × 2048, and the target size is 2/3 inch. It is equipped with Gigabit Ethernet, the lens is distortion free, the focal length is 8 mm, and the shooting effect is good under low illumination. Vs2019 and OpenCV were used as image processing software in the experiment. The process is completed on the computer with 8 GB memory and 2.50 GHz CPU working frequency.

### 5.1. Camera Calibration

The creel is in the three-dimensional world, and the creel photos that are taken are two-dimensional. If the camera is regarded as a function, the input is a scene, and the output is a gray-scale image [35]. In the process from three-dimensional to two-dimensional, the function is irreversible. The goal of camera calibration is to find a suitable mathematical model, then calculate the parameters of the model, approximately simulate the process from three-dimensional to two-dimensional, and finally find the inverse function [36,37]. Therefore, camera calibration is one of the basic works of visual measurement and positioning. The accuracy of calibration parameters is directly related to the accuracy of creel. Generally, camera calibration methods are divided into four methods: traditional camera calibration method, active vision camera calibration method, camera self calibration method and zero distortion camera calibration method.

Zhang Zhengyou calibration method is adopted in this paper. Strictly speaking, Zhang Zhengyou calibration method is between the traditional calibration method and the self calibration method. The GML camera calibration toolbox is used as the calibration software, in which the number of calibration images is 20, and the checkerboard specification is 10-11-14 mm printing paper. The camera calibration effect is shown in Figure 10. The observation shows that the corners of the black-and-white checkerboard in the image are accurately identified.

The specific camera parameters obtained after calibration are shown in Figure 11, key parameters are as follows. Focal Length (Focal length in X, Y axis direction $f_{x}$, $f_{y}$) is [2317.659 2314.165] ± [3.266 3.275]. Principal Point ($u_{0}$, $v_{0}$) is [1242.338 975.647] ± [2.490 2.656]. Distortion (Lens distortion coefficient) is [−0.149059 0.227607 −0.000687 −0.001450] ± [ 0.009618 0.130363 0.000270 0.000264]. The Camera Matrix is shown as Formula (10).
(10)$$M=\left[\begin{array}{c} f_{x}\\ 0\\ 0 \end{array}\begin{array}{c} 0\\ \;f_{y}\;\;\;\\ 0 \end{array}\begin{array}{c} u_{0}\\ v_{0}\\ 1 \end{array}\right]=\left[\begin{array}{c} 2317.659\\ 0\\ 0 \end{array}\begin{array}{c} 0\\ 2314.165\;\;\;\;\\ 0 \end{array}\begin{array}{c} 1242.338\\ 975.647\\ 1 \end{array}\right]$$

### 5.2. Image Processing and Positioning

After the camera calibration was completed, the calibrated camera parameters were directly inputted into the improved circle detection program of OpenCV platform to process and recognize the creel image, and automatically correct the positioning data according to the calibrated camera parameters.

In order to ensure the positioning accuracy of the creel and prevent interference with the yarn frame, it is necessary to determine the optimal distance for camera positioning. We selected the distance between the camera lens center and the marker center 160 mm, 170 mm, 180 mm, 190 mm, 200 mm and 210 mm, respectively, then conducted a large number of image processing at these six distances. We randomly select one image of each interval, as shown in Figure 12a–f.

According to the image processing results, marker diameter, pixel size and other parameters, using the pinhole imaging principle, the pixel can be converted to the positioning distance through similar triangle calculation. The diameter of the marker is 10 mm, and the pixel size can be obtained according to Formula (7) or read according to the camera manual. Finally, the pixel size was 3.45 um × 3.45 um.
Pixel size = target size/total pixel size(11)

### 5.3. Analysis of Experimental Data

We selected six intervals of 160 mm, 170 mm, 180 mm, 190 mm, 200 mm and 210 mm, respectively, for testing. In order to ensure the accuracy of the positioning data, in the experimental test, 10 images were quickly collected each time as a group, and then the 10 images were processed to obtain the 10 data, and the average value of the 10 data was obtained as one data. A total of 13 sets of images were collected in the experiment, and finally 13 data were obtained. After calculation, 13 groups of data at six intervals were drawn into curves, which are shown in Figure 13.

It can be seen from the Figure 13 that the maximum errors of the five spacing of 160 mm, 170 mm, 180 mm, 190 mm, 200 mm and 210 mm are 6.49 mm, 3.61 mm, 3.37 mm, 1.47 mm, 1.77 mm and 4.25 mm, respectively. So, we can conclude that only 190 mm and 200 mm meet the requirements if the bobbin yarn replacement accuracy is 2 mm.

After the 13 groups of data measured for each interval in Figure 10 are averaged, the results are shown in Table 1.

It can be seen from the Table 1: (1) When the spacing is 190 mm, the average error measured is the smallest, only 0.51 mm, and the relative error is 0.27%, which is far less than the accuracy requirements of 2 mm; (2) The average error of spacing of 180 mm and 200 mm is only 1.17 mm and 0.68 mm, which also meets the requirements; (3) The average errors of 160 mm and 210 mm spacing reached 5.12 mm and 4.88 mm, respectively, which exceeded the accuracy requirements and did not meet the requirements of creel positioning.

Based on the conclusions in Figure 10 and Table 1, the distance between the camera lens center and the center of the marker is an important factor to ensure the positioning accuracy of the creel. When using this monocular vision system to locate the creel, the best spacing range of image shooting is 190 mm~200 mm. Therefore, the positioning method proposed in this paper has a wide range of spacing, which reduces the difficulty of installation and debugging of the positioning system, and it is helpful to promote this method to practical production and application. In addition, the yarn frames in the production workshop of the circular weft knitting machine are in a free state, the impact and touch in the daily production process may cause the overall displacement of the yarn frames. If the distance between the camera lens center and the marker center is set to be 190 mm in the program, and the yarn frame displacement distance is less than 10 mm, it will not have a great impact on the automatic bobbin yarn changing, otherwise manual intervention will be taken, and a lot of labor resources will be occupied. Therefore, in the automatic bobbin yarn changing system, it is recommended to fix the bottom of the yarn frame with the ground.

In order to analyze the cause of the positioning error and further reduce the error, the distance between the camera lens center and the center of the marker is adjusted to 190 mm for testing. During the test, we take the end face of the marker as the coordinate plane, where x, y and r are the X-direction coordinates, Y-direction coordinates and the radius of the end face of the marker on the plane, respectively. Since it is impossible to precisely adjust the optical center of the camera and the center of the end face of the marker on the same straight line. First, we locate the camera position and assume that the center coordinates of the end face of the marker are (0, 0), and the radius r is 5 mm through measurement. Then, we use the improved Hough circle detection algorithm to identify the end face of the marker and calculate the coordinates and r corresponding to the center of the circle through the small hole imaging principle. The obtained data and their average values are shown in Table 2.

It can be seen from the Table 2 that the deviation between the preliminary results of x, y and r, and the positioning results is very small. The maximum deviation in X direction is −0.310 mm, and the average deviation is −0.3036 mm. The maximum deviation in Y direction is 0.889 mm, and the average deviation is 0.6395 mm. The average value of radius r is 4.794 mm, and the average deviation is −0.206 mm. The reason for the deviation may be the inaccurate detection of the end circle, and the center of the circle is not on the optical axis, etc., but the deviation is very small, which will not affect the practical application. In order to ensure the reliability of long-term operation of the system, the creel position can be calibrated regularly according to the actual situation.

In order to further test the reliability of the positioning system, we fine tune the positions of the camera and the creel according to the results obtained in Table 2. In this way, the optical center of the camera and the center of the end face of the marker shall be in a straight line as far as possible. Then we test the X-direction offset of −20 mm, −10 mm, 0 mm, 10 mm, 20 mm at the distance of 190 mm. We conclude that the left offset is negative and the right offset is positive. The specific data are shown in Table 3. Since the Y direction is similar, it will not be described in detail here.

It can be seen from Table 3 that when the right offset is 20 mm, the maximum positioning error is 1.11 mm, but it is still less than the accuracy requirement of 2 mm. In addition, after the camera position is adjusted, the coordinate deviation of the center of the circle becomes smaller, but it has not been completely eliminated. However, it does not affect the actual use.

To sum up, it is completely feasible to use the end face of the marker as the positioning marker of the creel at a spacing of 190 mm, and the positioning error is very small. When in use, the deviation data is imported into the controller through the Ethernet bus to correct the deviation, and the bobbin automatic yarn change is completed by the bobbin yarn change manipulator according to the updated data movement. At the same time, the time used for image analysis, positioning processing and data communication of the creel was counted during the experiment. The time used for positioning was within 1 s, which could meet the time requirements of creel positioning calculation when the bobbin yarn changing manipulator was working.

## 6. Conclusions and Discussion

### 6.1. Conclusions

Aiming at the problem that the coordinate of the bobbin needs to be initialized regularly when the bobbin is automatically changed on the bobbin by the manipulator in the textile production workshop, this paper proposes a bobbin positioning technology based on monocular vision. Through the design of creel identification markers, the improved Hough circle detection algorithm was used for identification and positioning. This method reduces the difficulty of creel positioning, recognition and the requirements of processor configuration. At the same time, the positioning results and detection time can meet the requirements of use. Meanwhile, it has the advantages of low cost, easy installation and easy industrialization. So, it can provide a certain basis for the industrial application of automatic bobbin change in circular weft frames.

### 6.2. Limination

Since there are many rows of yarn frames in the production workshop of circular weft machine, the background is complex; however, we reduce the difficulty of identification of creels by using markers and realize the positioning of creels in principle. However, due to the small area of the end face of the creel, the vibration of the manipulator during operation has a great impact on the accuracy and stability of the identification. Therefore, it is necessary to shoot after the camera is really stopped. In addition, there are many yarn racks in the yarn area. The current identification method can only locate a single creel at one time. So, there is still a problem of low positioning efficiency for practical application.

### 6.3. Discussion

Future work will mainly improve the positioning efficiency and accuracy of the system from two aspects. On the one hand, starting with the performance of hardware equipment, we choose industrial computers with a higher resolution and faster processing capacity to overcome the problems of small end area of creel and high recognition accuracy. On the other hand, starting from the recognition and positioning algorithm, machine learning or deep learning is used to improve the accuracy and effect of positioning, realize the recognition of multiple creels at a single time, and promote the practical application of this technology in textile workshops.

## Figures and Tables

**Figure 1 sensors-22-06657-f001:**
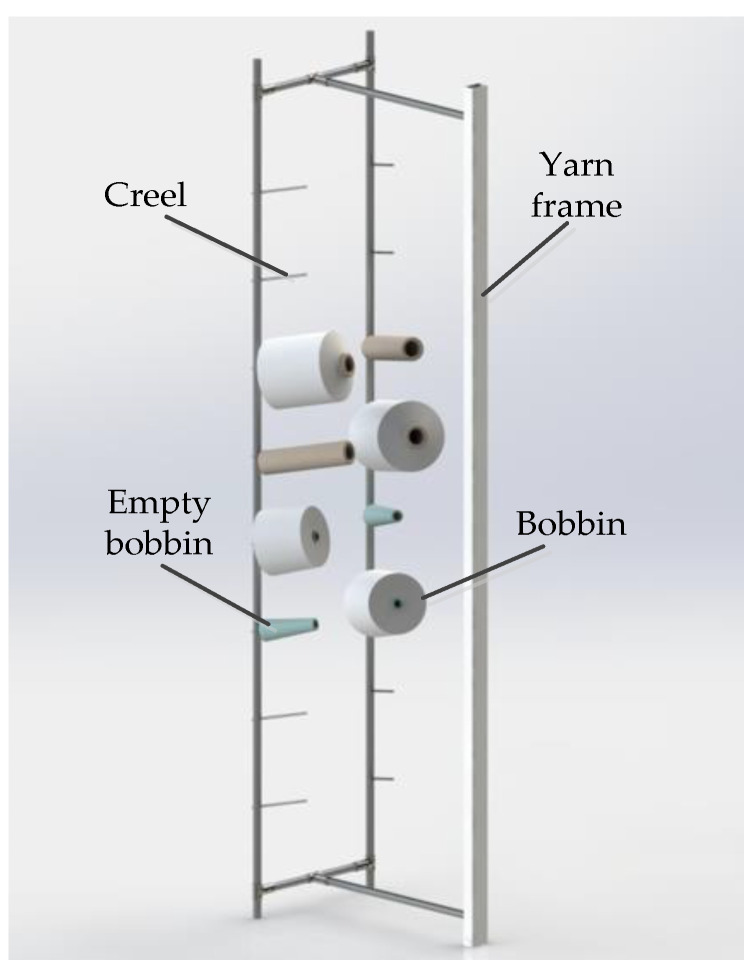
The structure diagram of the commonly used floor frame.

**Figure 2 sensors-22-06657-f002:**
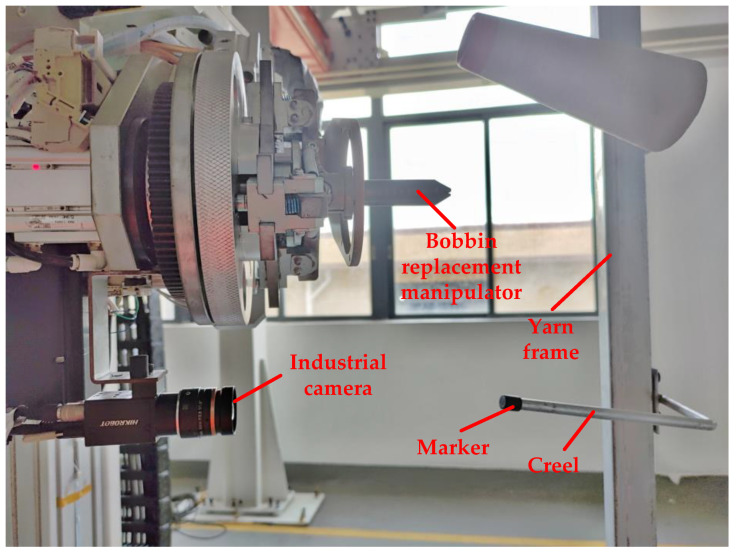
This is the physical drawing of creel visual positioning system.

**Figure 3 sensors-22-06657-f003:**
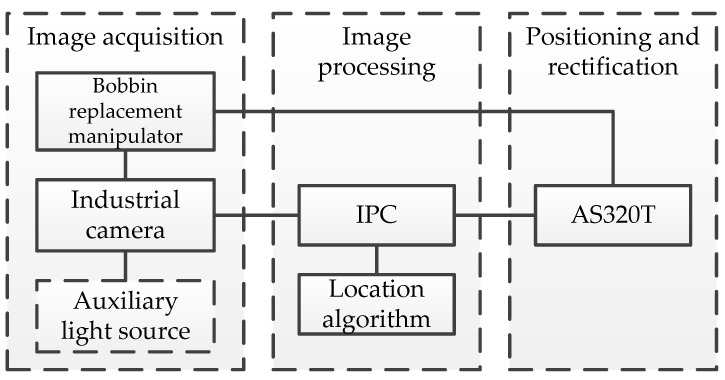
This is the composition block diagram of creel positioning system.

**Figure 4 sensors-22-06657-f004:**
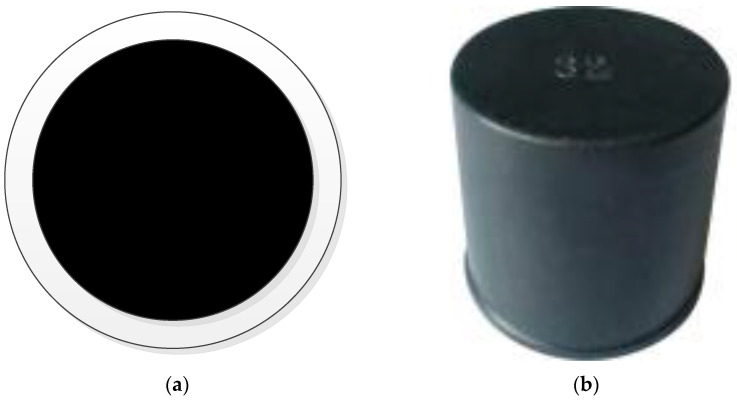
Shows the visual location markers. (**a**) Disc marker; (**b**) Cover marker.

**Figure 5 sensors-22-06657-f005:**
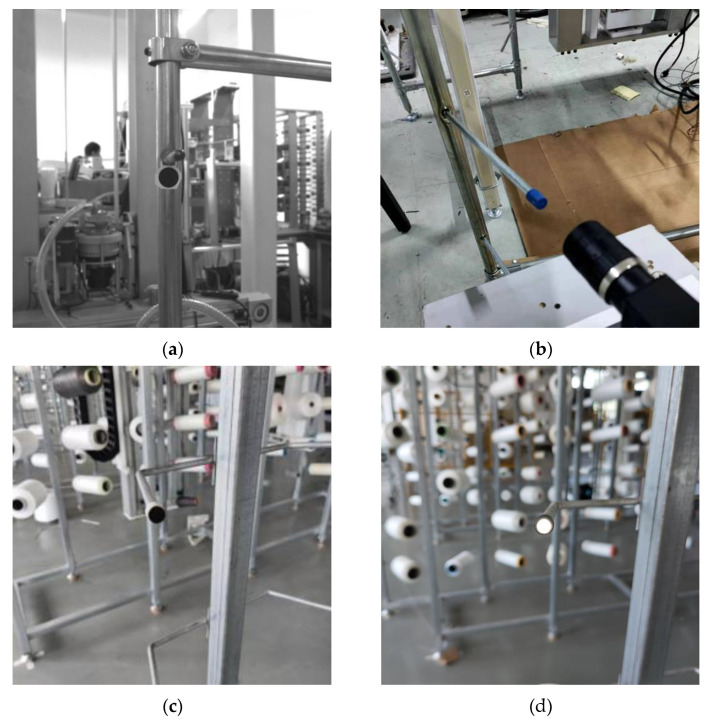
Shows the installation drawing of markers. (**a**) Physical installation drawing of disc marker; (**b**) Physical installation drawing of blue surface cover; (**c**) Physical installation drawing of black surface cover; (**d**) Physical installation drawing of reflective surface cover.

**Figure 6 sensors-22-06657-f006:**
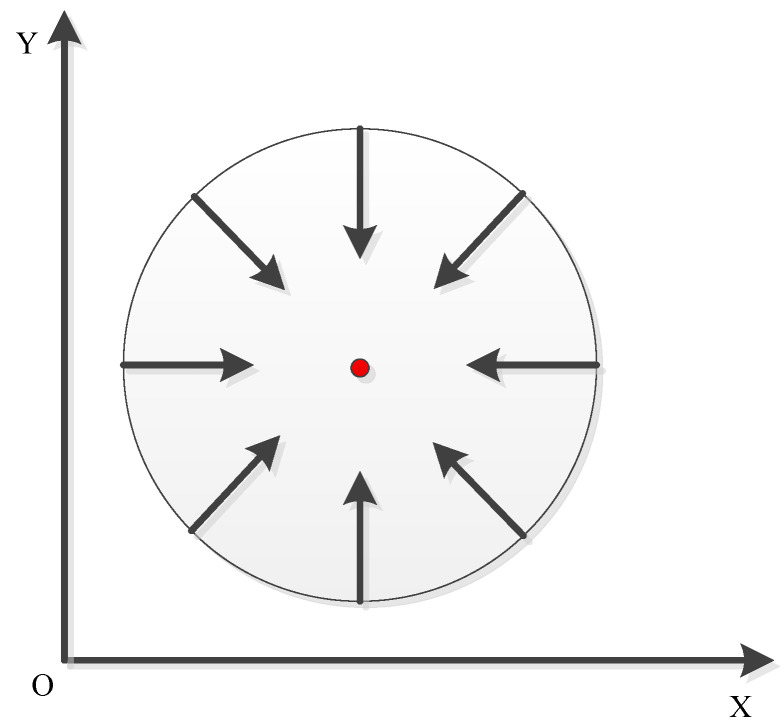
Schematic diagram of the relationship between module vector and circle center.

**Figure 7 sensors-22-06657-f007:**
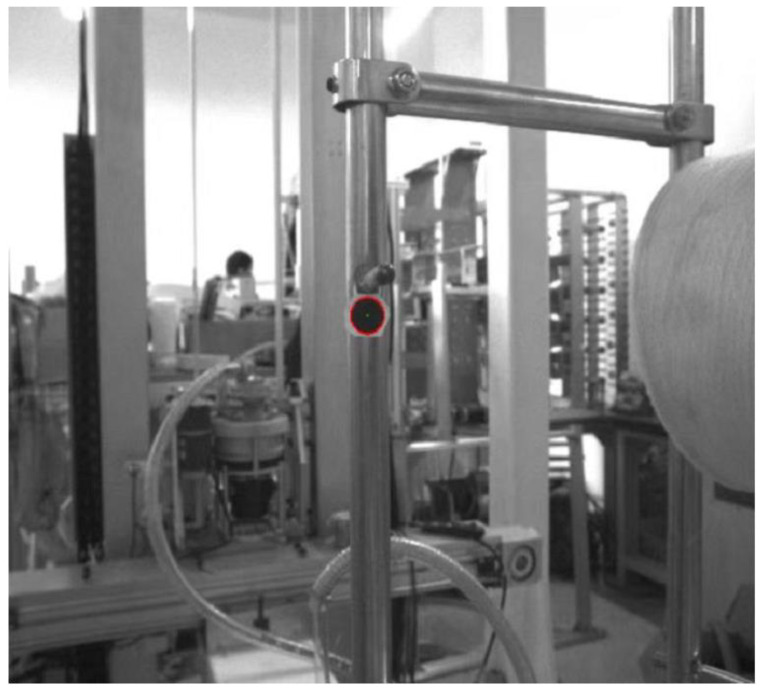
Effect drawing of disc marker recognition.

**Figure 8 sensors-22-06657-f008:**
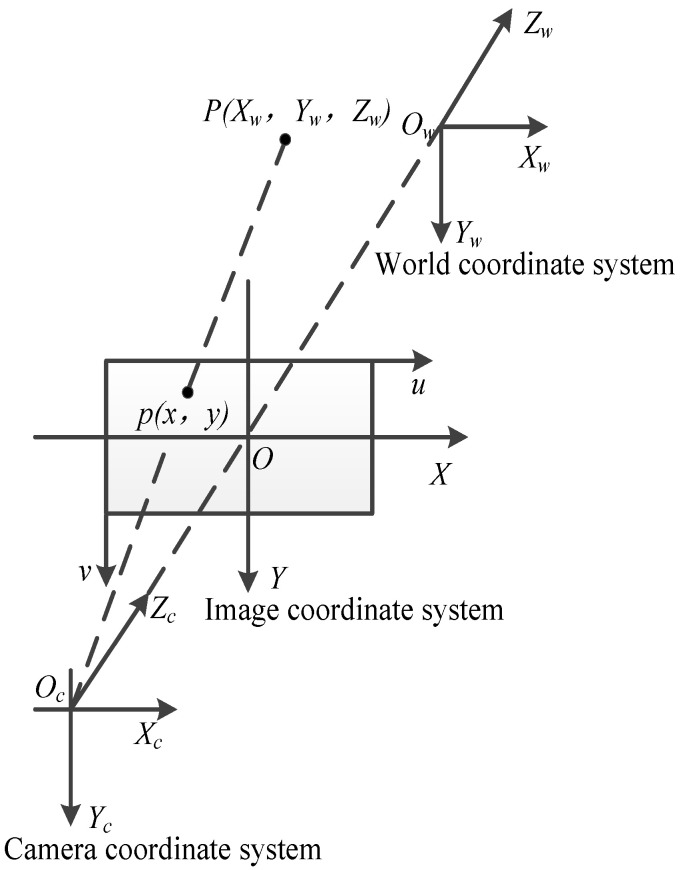
Creel positioning schematic diagram.

**Figure 9 sensors-22-06657-f009:**
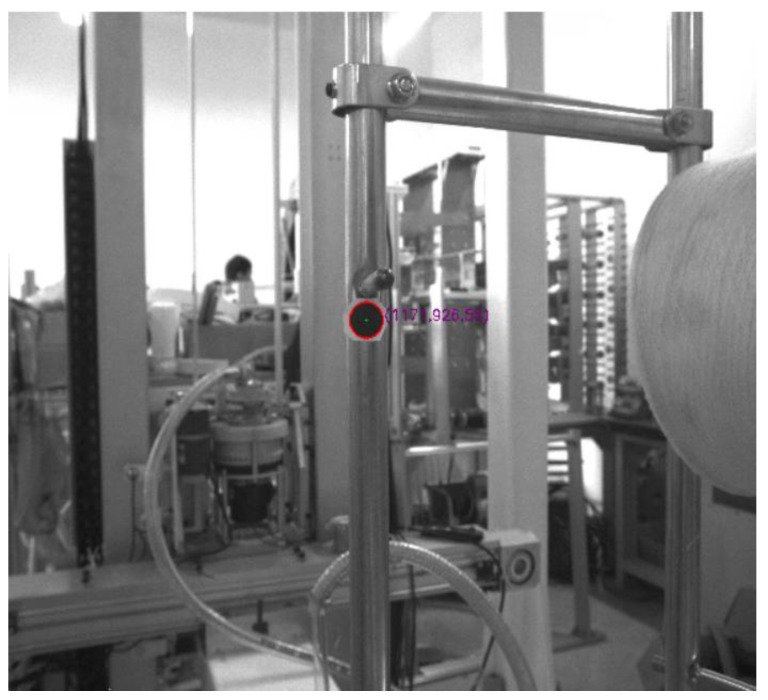
Effect drawing of creel positioning.

**Figure 10 sensors-22-06657-f010:**
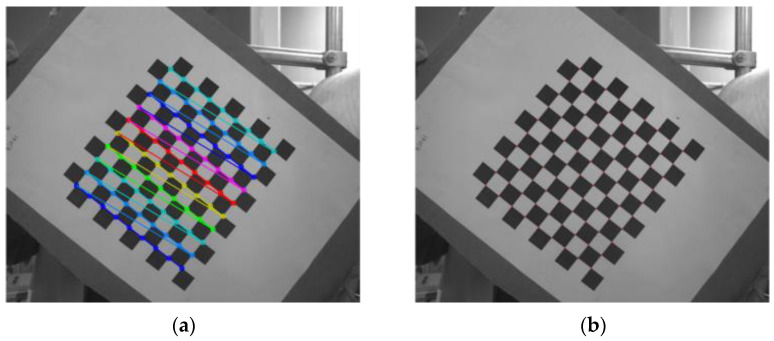
Camera calibration diagram. (**a**) Calibration object; (**b**) Reproject.

**Figure 11 sensors-22-06657-f011:**
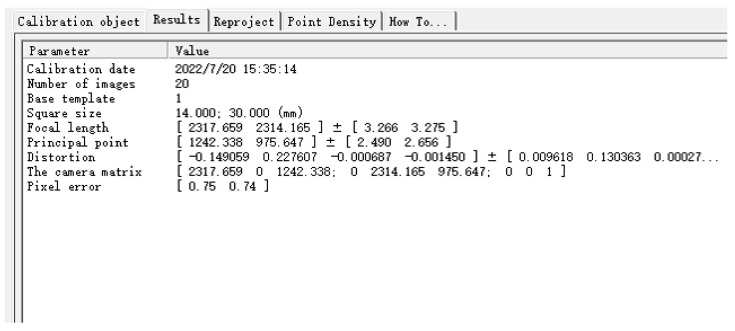
Camera calibration result diagram.

**Figure 12 sensors-22-06657-f012:**
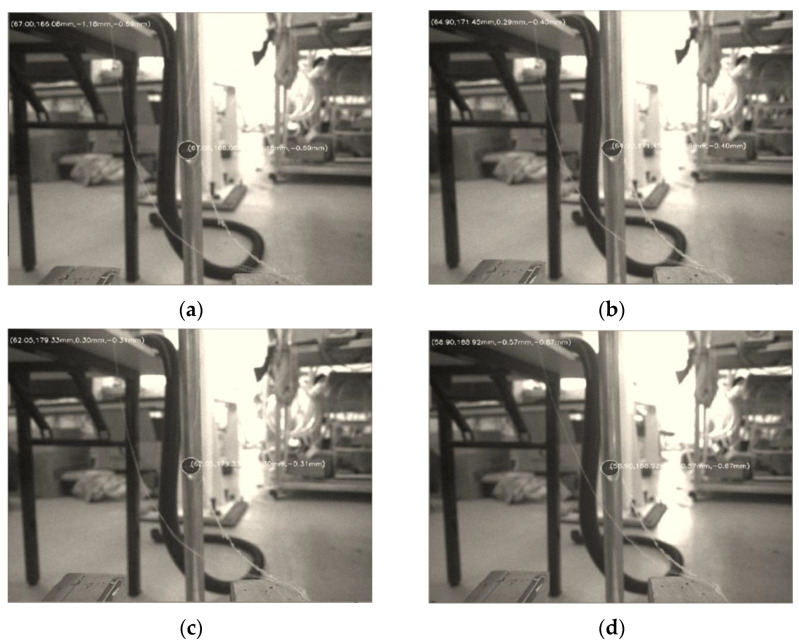
Image processing results of each interval. (**a**) Image processing results of 160 mm; (**b**) Image processing results of 170 mm; (**c**) Image processing results of 180 mm; (**d**) Image processing results of 190 mm; (**e**) Image processing results of 200 mm; (**f**) Image processing results of 210 mm.

**Figure 13 sensors-22-06657-f013:**
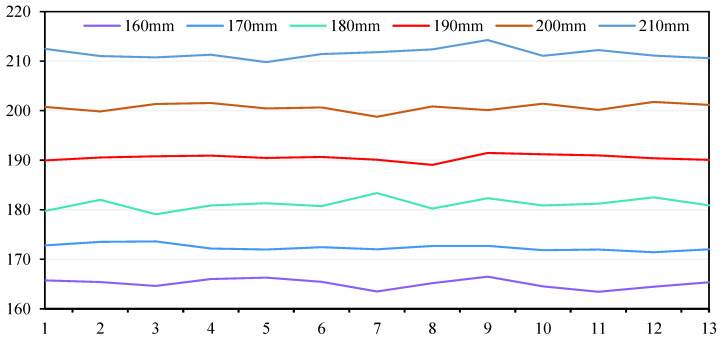
Image processing data curve of each interval.

**Table 1 sensors-22-06657-t001:** This table shows the creel positioning results under different test spacing.

Measured Spacing/mm	Positioning Spacing/mm	Absolute Error/mm	Relative Error
160	165.12	5.12	3.20%
170	172.40	2.40	1.41%
180	181.17	1.17	0.65%
190	190.51	0.51	0.27%
200	200.677	0.677	0.34%
210	224.88	4.88	2.21%

**Table 2 sensors-22-06657-t002:** This table shows the creel positioning data at 190 mm spacing.

Preliminary Result/mm (x, y, r)	Positioning Result/mm (x’, y’, r’)	Deviation/mm (∆x, ∆y, ∆r)
0, 0, 5	−0.304, 0.889, 4.83	−0.304, 0.889, −0.17
0, 0, 5	−0.310, 0.685, 4.77	−0.310, 0.685, −0.23
0, 0, 5	−0.308, 0.681, 4.81	−0.308, 0.681, −0.19
0, 0, 5	−0.296, 0.443, 4.85	−0.296, 0.443, −0.15
0, 0, 5	−0.298, 0.445, 4.86	−0.298, 0.445, −0.14
0, 0, 5	−0.310, 0.685, 4.79	−0.310, 0.685, −0.21
0, 0, 5	−0.301, 0.557, 4.75	−0.301, 0.557, −0.25
0, 0, 5	−0.304, 0.564, 4.77	−0.304, 0.564, −0.23
0, 0, 5	−0.303, 0.561, 4.75	−0.303, 0.561, −0.25
0, 0, 5	−0.302, 0.885, 4.76	−0.302, 0.885, −0.24
0, 0, 5	−0.3036, 0.6395, 4.794	−0.3036, 0.6395, −0.206

**Table 3 sensors-22-06657-t003:** This table shows the positioning results under different offset distances.

Measured Result/mm (x, y)	Positioning Result/mm (x’, y’)	Deviation/mm (∆x, ∆y)
−20, 0	−19.69, 0.50	0.31, 0.50
−10, 0	−9.85, 0.46	0.15, 0.46
0, 0	0.26, 0.51	0.26, 0.48
10, 0	10.81, 0.40	0.81, 0.40
20, 0	21.11, 0.38	1.11, 0.38

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
