# Peer review of "The Method of Creel Positioning Based on Monocular Vision"

_sensors, 2022, doi:10.3390/s22176657_

Round 1

Reviewer 1 Report

Great work, ready to be published as it is. It is presented through the design of creel identification markers, the improved  Hough circle detection algorithm is used for identification and positioning.

Author Response

Dear commentator:

Thank you for your decision and constructive comments on my manuscript. We have carefully considered your suggestions and made some modifications. We have tried our best to improve and made some changes to the manuscript.

Reviewer 2 Report

The authors presented an article devoted to the automation of the process of changing bobbins on a knitting machine using creel positioning through computer vision methods. The article practically does not contain elements of scientific novelty, but is a solution to a specific applied problem, which is relevant for the textile industry. From a technical point of view, the presented description of the methods used is sufficient, and somewhere even redundant. The reviewer has a few questions and comments:

1. In formulas (2), (3) it is required to explain what theta is. It is also needed to add punctuation marks (this remark applies to all formulas).

2. On page 8, the description of the Hough algorithm for detecting a circle seems redundant, especially since the authors took a ready-made solution in the opencv library.

3. In formulas (7), (8), match the designation of quantities in the formulas and in the text after the formulas (indices, case)

Author Response

Dear commentator:

Thank you for your decision and constructive comments on my manuscript. The focus of this paper is really the application of machine vision technology in textile intelligent manufacturing process. At present, the replacement of the bobbin of the yarn frame is still purely manual, which is not in line with the direction of textile manufacturing. Therefore, we want to use the existing technology to replace labor, and have done some work in this regard, and hope to do something for the realization of automatic replacement of bobbins. We have carefully considered your suggestions and made some modifications. We have tried our best to improve and made some changes to the manuscript. The red part is modified according to your comments. The description of point-to-point revision is as follows:

  1. In formulas (2), (3) it is required to explain what theta is. It is also needed to add punctuation marks (this remark applies to all formulas).

In formulas (1): (x, y) is the point on the circumference.

In formulas (2), (3): θ is the angle between the line connecting the center (a, b) and the point (x, y) on the circle and the horizontal line passing through the center (a, b).

  1. On page 8, the description of the Hough algorithm for detecting a circle seems redundant, especially since the authors took a ready-made solution in the opencv library.

We consider this paragraph as follows: since the parameters in Hough circle detection algorithm need to be adjusted during OpenCV algorithm debugging, we want to explain the detection principle clearly. These contents are indeed superfluous for skilled scientific researchers. However, considering that some beginners are not familiar with the principle, if the topic is used as a learning case, it may require additional time to find relevant materials to help understand. Therefore, this paper describes it to a certain extent. If it is really unnecessary, I will delete it.

  1. In formulas (7), (8), match the designation of quantities in the formulas and in the text after the formulas (indices, case)

In formulas (8): υ0, ν0  is the actual position of the main point.

In formulas (8): R is the rotation matrix and T is the translation matrix.

Reviewer 3 Report

An interesting article, but mostly of the practical importance. The research analysis supported by an appropriate literature review. Clearly and well-described research method.

Critical comments to the article:

1. The article shoudl be written in the passive voice.

2. The ending is missing in the first sentence of section 3.2: "Because the diameter of the end face of the creel is about 8mm, the color is silver white, and the workmanship is slightly rough." Because what?

3. Figure 9 is illegible. What's different from Figure 7. The purple text in Figure 9 is illegible.

4. Why was this camera calibration method chosen? There is no justification.

5. Why there were the 13 groups of results created? Later in Table 1 all results are already averaged. It needs to be explained. If these groups are important for some reason, how were 10 images selected for the group - randomly?

Author Response

Dear commentator:

Thank you for your decision and constructive comments on my manuscript. This paper is really biased towards practical application, because the bobbin replacement of the yarn frame is still purely manual, which is not in line with the direction of textile manufacturing. Therefore, we want to use the existing technology to replace the labor force and have done some work in this regard, and hope to do something for the realization of automatic replacement of bobbins.. We have carefully considered your suggestions and made some modifications. We have tried our best to improve and made some changes to the manuscript.The purple part is modified according to your comments. The description of point-to-point revision is as follows:

  1. The article should be written in the passive voice.

Authors have carefully read the full text and revised the problems we found. There may still be some problems, please do not hesitate to comment. Thank you!

  1. The ending is missing in the first sentence of section 3.2: "Because the diameter of the end face of the creel is about 8mm, the color is silver white, and the workmanship is slightly rough." Because what?

What I want to express here is that the end face of the bobbin has problems such as small area, rough indication, color reflection, etc., which is the reason for adding markers. Maybe we didn't express clearly here. 

We revised it as follows on page 6: There are the following problems in image recognition of the end face of the bobbin directly: the diameter of the end face is only about 8mm, the color is silver white, and the process is slightly rough.

  1. Figure 9 is illegible. What's different from Figure 7. The purple text in Figure 9 is illegible.

Figure 9 is indeed somewhat illegible, we replaced Figure 9 with a clearer figure.

Figure 7 shows the identification of circles.

Figure 9 shows the positioning effect of marking the center of the circle, where the green point is the center position, and also shows the center coordinates and radius values.

  1. Why was this camera calibration method chosen? There is no justification.

I'm very sorry. I may have made a mistake here. Zhang Zhengyou calibration method is used in this paper. Strictly speaking, it is between the traditional calibration method and the self calibration method. It is not the self calibration method as described in the original text.

  1. Why there were the 13 groups of results created? Later in Table 1 all results are already averaged. It needs to be explained. If these groups are important for some reason, how were 10 images selected for the group - randomly?

Maybe we didn't express clearly here. In order to ensure the accuracy of the positioning data, in the experimental test, 10 images were quickly collected each time as a group, and then the 10 images were processed to obtain the 10 data, and the average value of the 10 data was obtained as one data. A total of 13 sets of images were collected in the experiment, and finally 13 data were obtained, as shown in Figure 13.

The data in Table 1 was obtained by taking the average value of 13 groups of data in Figure 13, in order to analyze the test conditions at different intervals.